# Joint analysis of phenotype-effect-generation identifies loci associated with grain quality traits in rice hybrids

Lanzhi Li [1,8], Xingfei Zheng[2,3,8], Jiabo Wang[4,8], Xueli Zhang[1], Xiaogang He[1], Liwen Xiong[1], Shufeng Song [5], Jing Su[1], Ying Diao[6], Zheming Yuan[1], Zhiwu Zhang [7] ✉ & Zhongli Hu [3,6] ✉

Genetic improvement of grain quality is more challenging in hybrid rice than in inbred rice due to additional nonadditive effects such as dominance. Here, we describe a pipeline developed for joint analysis of phenotypes, effects, and generations (JPEG). As a demonstration, we analyze 12 grain quality traits of 113 inbred lines (male parents), five tester lines (female parents), and 565 (113×5) of their hybrids. We sequence the parents for single nucleotide polymorphisms calling and infer the genotypes of the hybrids. Genome-wide association studies with JPEG identify 128 loci associated with at least one of the 12 traits, including 44, 97, and 13 loci with additive effects, dominant effects, and both additive and dominant effects, respectively. These loci together explain more than 30% of the genetic variation in hybrid performance for each of the traits. The JEPG statistical pipeline can help to identify superior crosses for breeding rice hybrids with improved grain quality.

More than half of the world's population consumes rice (*Oryza sativa* L.) as their staple food. With the improvements in living standards, consumers are paying more attention to rice end-use cooking quality[1]. Many rice quality traits, including protein content and chalkiness degree, are correlated with yield[1]. The yields of many rice hybrids are higher than those of conventional inbred varieties, but the grain quality of hybrids often needs improvement. To obtain both high-quality and high-yield rice hybrids, the quality traits of hybrids need to be thoroughly genetically dissected in order to identify superior crosses[2,3].

Rice grain quality has been classified into milling, appearance, cooking and eating, and nutritional categories[1,4]. The rice milling quality determines the yield and appearance of rice after the milling process. Milling quality comprises the brown rice ratio, milled rice ratio, and head rice ratio (BRR, MRR, and HRR, respectively). A large HRR is one of the most important criteria for measuring milled rice quality. Appearance quality refers to how rice appears after milling and is associated with grain length (GL), grain width (GW), the grain length–width ratio (GLWR), and translucency/chalkiness of the endosperm. Most consumers prefer translucent rice as opposed to chalky rice. Cooking and eating quality include the easiness of cooking as well as the texture, springiness, stickiness, and chewiness of cooked rice, factors that are controlled by starch's physical and chemical properties and involve amylose content and alkali spreading value. The amylose content (AC) of rice is known to play a crucial role in determining its cooked texture. The alkali spreading value (ASV) is a standard assay

[1]Hunan Engineering & Technology Research Center for Agricultural Big Data Analysis & Decision-making, College of Plant Protection, Hunan Agricultural University, 410128 Changsha, Hunan, China. [2]Hubei Key Laboratory of Food Crop Germplasm and Genetic Improvement, Food Crop Institute, Hubei Academy of Agricultural Sciences, 430064 Wuhan, Hubei, China. [3]State Key Laboratory of Hybrid Rice, College of Life Science, Wuhan University, 430072 Wuhan, Hubei, China. [4]Key Laboratory of Qinghai-Tibetan Plateau Animal Genetic Resource Reservation and Utilization of Ministry of Education and Sichuan province, Southwest Minzu University, 610041 Chengdu, Sichuan, China. [5]State Key Laboratory of Hybrid Rice, Hunan Hybrid Rice Research Center, Hunan Academy of Agricultural Sciences, 410125 Changsha, Hunan, China. [6]School of Life Science and Technology, Wuhan Polytechnic University, 430023 Wuhan, Hubei, China. [7]Department of Crop and Soil Sciences, Washington State University, Pullman, WA 99164, USA. [8]These authors contributed equally: Lanzhi Li, Xingfei Zheng, Jiabo Wang. ✉e-mail: Zhiwu.Zhang@WSU.edu; huzhongli@whu.edu.cn

used to classify processing and cooking quality; it provides a simple means of classifying rice into high, intermediate, and low gelatinization temperature types. Protein content (PC) is a major index of rice grain nutritional quality. Since storage protein affects rice texture and processing quality, an intermediate PC is generally preferred[4].

Many rice grain quality traits in inbred varieties have been well studied. Multiple genes have been cloned and localized, and their mechanisms and functions have been investigated. Grain size is closely related to yield[4,5]. At present, dozens of grain size-related genes have been isolated from multiple rice germplasm resources, including the genes *GS3*, *GL3.1*, and *GW7/GL7* that control grain length[6–8], the genes *GW2*, *GW5/qSW5*, and *GS5* that control grain width[9–11], and the genes *GS6*, *GS9*, *TGW6*, and *GW8*/SPL16 that control grain size[5,12–14]. Chalky grains are considered low quality because of their poor appearance and undesirable cooking and milling qualities[15]. The rice gene *OsRab5a* regulates endomembrane organization and storage protein trafficking in rice endosperm cells, factors that affect the formation of amyloplasts[16]. *Chalk5* encodes a vacuolar H + -translocating pyrophosphatase that influences grain chalkiness in rice[17].

The amylose content (AC) has the greatest influence on the cooking and consumption qualities. The synthesis of rice amylose is catalysed by a granule-bound starch synthase protein encoded by the genes *Waxy* and *Wx*[18]. There are several alleles of the *Wx* gene, including $Wx^a$, $Wx^b$, $wx$, $Wx^{mp}$, $Wx^{op}$, $Wx^{in}$, and $Wx^{mq}$. Zhou et al.[19] cloned a defective soluble starch synthase gene (*SSIIIa*) responsible for resistant starch production and further showed that production depends on high expression of the $Wx^a$ allele. Rice varieties with an intermediate gel temperature, which is predominantly determined by the amylopectin structure, are generally preferred by consumers. The gene (chr06:6748398_6753302 (+ strand)), starch synthase II (*OsSSIIa*), is the major determinant of gel temperature[4].

In breeding programs, the selection of parental inbred lines for the development of superior hybrids is challenging[20]. Genetic dissection of hybrids is more difficult than for inbred lines, as nonadditive genetic effects such as dominance are involved in addition to additive genetic effects. Furthermore, the joint analysis of these genetic effects requires the integration of both inbred and hybrid populations. In many analyses, heterosis is estimated as the difference between the hybrid and mid-parent values from parent–child trios to map loci associated with dominant effects. The dominant effects should be treated in both directions according to the reference allele. For example, heterozygous genotypes are coded to be homozygous genotypes of the reference allele in the dominant model and homozygous genotypes of the alternative allele in the recessive model[21].

Joint additive and dominant effect models have demonstrated superiority over models with separate effects. A genome-wide association study (GWAS) of 130,725 cattle using a joint additive and nonadditive model identified six dominant loci with impacts exceeding the largest effect variant identified by the corresponding additive effect model[22]. When both hybrid and inbred parent populations are available, the differences and similarities among parental inbred phenotypes, hybrid phenotypes, general combining ability, and hybrid heterosis can be used to infer genetic effects. In maize, 1428 maternal inbred varieties were crossed with 30 paternal inbred varieties to generate 42,840 (1428 × 30) hybrids. There were 166 quantitative trait loci (QTLs) identified for three agronomic traits: days to tasselling, plant height, and ear weight. These QTLs were categorized into three classes (additive, dominant, or epistatic effects)[23] using comparisons to models with a single effect in a single population. Ideally, both additive and dominant genetic effects should be analysed simultaneously with both parental inbred and hybrid populations to maximize statistical power.

In this study, we cross 113 male inbred varieties with five female inbred parents and generated 565 (113×5) hybrid test crosses. Both parental inbred varieties (*V*) and hybrid test crosses (*T*) were scored for

phenotype for 12 quality traits. The parental inbred varieties are genotyped using whole-genome sequencing. The genotypes of hybrids are inferred from the genotypes of their parents. General combining ability (*G*) and heterosis (*H*) are derived for maternal inbred progeny and hybrids, respectively. We develop a statistical model and a computing pipeline to simultaneously analyse additive and dominant genetic effects using both original phenotypes (*V* and *T*) and derived phenotypes (*G* and *H*) from parental inbreds and hybrids, and identify 128 genetic loci associated with additive and/or dominant genetic effects on the 12 grain quality traits. These loci together explain more than 30% of the genetic variation in hybrid performance for each of the traits.

## Results

### Rice grain quality traits in inbred lines and hybrids

Most of the 12 rice quality traits had normal distributions in inbred varieties, including PC, GW, MRR, and BRR (Supplementary Fig. 1). There were two traits that showed skewed distributions (ASV towards the lower end and percentage of grains with chalkiness (PGWC) towards the upper end). High ASV and low PGWC are preferred by customers. There were also traits with a bimodal distribution, including chalkiness degree (CD) and AC. Low CD and AC are preferred by customers[1]. The deviation from normality revealed the selection for quality improvement. Test crosses had similar distributions with peaks at different locations for all traits except transparent degree (TD). The three peaks were more pronounced in test crosses than the peaks in inbred lines. High TD is preferred by customers; unfortunately, the five female parent inbred varieties were below average compared to the male parent inbred varieties. Consequently, hybrids had undesirable TD values compared to their male parent inbred varieties.

In general, the values of the hybrids were between those of their male and female parents, suggesting an additive genetic effect and the critical role of selecting inbred varieties to improve hybrids. For example, all of the female parent inbred varieties were more desirable than the average male parent inbred varieties in terms of CD and PGWC. Consequently, the hybrids were more desirable than their male parents for these two traits. Similar phenomena were also observed for other traits. The higher the performance of the female parent, the higher the performance in the hybrids and the greater the differences between the hybrids and their male parent inbred varieties. However, the order of female parents did not remain constant in the hybrids, suggesting the complexity of hybrid quality traits (Supplementary Fig. 2). Taking PGWC as an example, the value of *Y58S* was higher than the value of *3A*. The median PGWC of hybrids parented by *Y58S* was lower than that of the hybrids parented by *3A*. HRR presented another example. The HRR of inbred *3A* was higher than that of *GZ63*. The median HRR of hybrids parented by *3A* was lower than that of the hybrids parented by *GZ63*.

### Relationships among quality traits

There were strong positive correlations between CD and PGWC for four phenotypes: *V* (0.93), *T* (0.92), *G* (0.93), and *H* (0.84) (Supplementary Fig. 3). There were no other traits that had such strong correlations with CD and PGWC. TD only had strong positive correlations with CD and PGWC in datasets *T* and *G*. PC and ASV were barely correlated with any other traits. These relationships suggest the possibility of improving PC and ASV simultaneously without affecting other traits.

Similar to studies reported in the literature[11,16], there were moderate negative correlations between GL and GW for four datasets, *V* (−0.5), *T* (−0.5), *G* (−0.57), and *H* (−0.42). GL had moderate negative correlations with CD and PGWC, while GW had moderate positive correlations with CD and PGWC. As a function of GL and GW, their ratio GLWR (GL/GW) had a strong positive correlation with GL and a strong negative correlation with GW. GL, GW, and GLWR had very weak

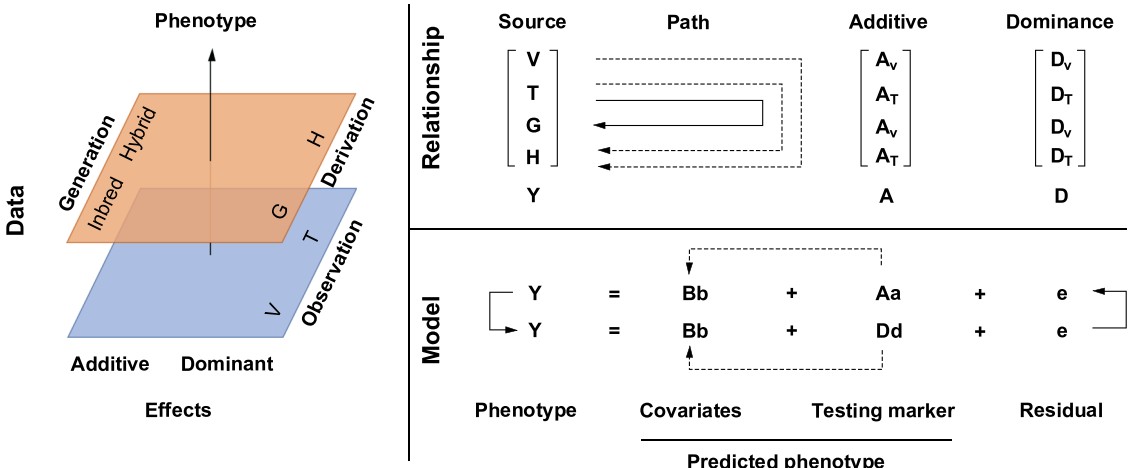

**Fig. 1 | Analysis pipeline of joint phenotypes, effects, and generations (JPEG).** The pipeline is summarized for phenotype source in the Data panel, phenotype derivation and the corresponding genotype in the Relationship panel, and decomposition of phenotypes into additive and dominant genetic effects in the Model panel. Original observations are measured for inbred varieties (*V*) and test crosses (*T*). The general combinability (*G*) of a variety was defined as the average performance of its testcross (the solid line in the relationship panel). Heterosis (*H*) of a test cross was defined as its difference between testcrosses and the middle parents, which uses both testcrosses and parents (the dashed lines in the relationship panel). Standardization is performed within each of *V*, *T*, *G*, and *H* to form the joint observations (Y). Additive genotypes (A) and dominant genotypes (D) are recorded or derived for the inbred variety (*V*) and test crosses (*T*), respectively (the data panel at the top). For an SNP, the additive genetic effect (a) is the regression performed on additive genotypes (A) and is coded as 0 and 2 for the two homozygotes and 1 for the heterozygotes. The dominant genetic effect (d) is the regression performed on dominant genotypes (D), which is coded as 0 for the two homozygotes and 1 for the heterozygotes. The JPEG pipeline can be conducted with multiple loci models for GWAS such as BLINK to iteratively add associated additive effects and dominant effects as covariates (the black arrows in the model panel at the bottom). At the end of each iteration, the associated A or D was added to the covariates (B) to control the population structure and cryptic association among individuals (the dashed lines from A/D to B). The initial covariates include the first three principal components derived from all markers and the dummy variables of female parents for the testcross. The iterations stopped when no additive or dominant effect could be added to covariates. At end of iteration, genomic prediction can be conducted by summing the additive and dominant genetic effects.

correlations with the remaining traits, especially AC, ASV, BRR, HRR, and MRR. These relationships should be considered during rice breeding for high yield and grain quality.

## Properties of single nucleotide polymorphisms
A total of 7,734,465 raw SNPs were obtained from 120 parental varieties genotyped by whole-genome sequencing with the Illumina HiSeq2500 platform. Among all the raw SNPs, 1,619,588 SNPs passed the quality control criteria of missing rate <20% and minor allele frequency (MAF) > 5%. Marker distributions were displayed as a heatmap for 12 chromosomes based on MAF (Supplementary Fig. 4A). Most (95%) intervals of adjacent SNPs were less than 200 kb. The average distance between markers was 196.8 bp. Approximately 58.88% of the distances between pairs of adjacent SNPs were less than 50 bp (Supplementary Fig. 4B). The MAF distribution was skewed to the low end, reflecting new genetic variants (Supplementary Fig. 4C). As a strict inbreeding species, rice has a low linkage disequilibrium (LD) decay rate. In our population, the rate of LD decay was not below an $r^2$ value of 0.3 on average within ~200 kb in the parent inbred varieties (Supplementary Fig. 4D).

## Population structure analyses
Most of the female parents (four out of five) were located at one end of the phylogenetic tree developed from the 1,619,588 filtered SNPs (Supplementary Fig. 5A). The others were in the middle of the tree, with the other end of the phylogenetic tree free of female parents; this creates the potential for heterosis. In general, the five female parents were genetically diverse. However, female parents *PA64* and *YS8S* were closely related. Similarly, *GZ63* and *AS* were very closely related. Similar phenomena were revealed by the principal component analysis (PCA) based on the 1,619,588 filtered SNPs. The first three principal components (PCs) explained 43.4% of the total variation in hybrid test crosses, compared to 27.6% in inbred varieties (Supplementary Fig. 6). The population structure in hybrid test crosses was stronger than in

inbred varieties. All of the female parents were near the midpoint of PC1 and the lower half of PC2 (Supplementary Fig. 6A–D). The principal component plots demonstrated the same findings as the phylogenetic tree. Female parents *AS* and *GZ63* were genetically similar, and the female parent *Y56S* was similar to *PA64*. Consequently, the hybrids of *Y56S* and the hybrids of *PA64* were closely related. The hybrids of *AS* and *GZ63* were completely scattered in the principal component plots (Supplementary Fig. 6E–H).

Four outliers were identified for test crosses within female parents. Their male parents were differentiated from the rest of the inbred varieties. These four inbred varieties had the lowest PC3 scores. *Nanjing11* had the lowest PC3 value, and *IRAT109* was almost identical to *Varylava* for all PCs, appearing as one heavy point. *Varylava* had the third lowest value of PC3. This variety is a type of early-season rice. *N22* had the fourth lowest value of PC3, with partial *japonica* components. All four outliers had foreign origins. The remaining 111 male parents were medium- or late-season *indica* rice varieties. The subpopulation structure was investigated using ADMIXTURE software (Supplementary Fig. 5B and C). The proportions belonging to subpopulations are displayed for inbred varieties sorted according to the proportion of the first subpopulation. The cross-validation demonstrated that $K = 5$ produced the minimum error, suggesting that five subpopulations should be recognized.

## Gene mapping with joint analysis of phenotypes, effects, and generations
We developed a pipeline of genome wide association study for a joint analysis of phenotypes, effects, and generations (JPEG) (Fig. 1). The pipeline can analyse both additive and dominant genetic effects and address different combinations of data sources, including parents, hybrids, and their combination. The phenotype data are vertically combined into a single vector. Both additive and dominant genotypes are required, whether the corresponding phenotype contains the genetic effect or not, to fit the computational requirements using

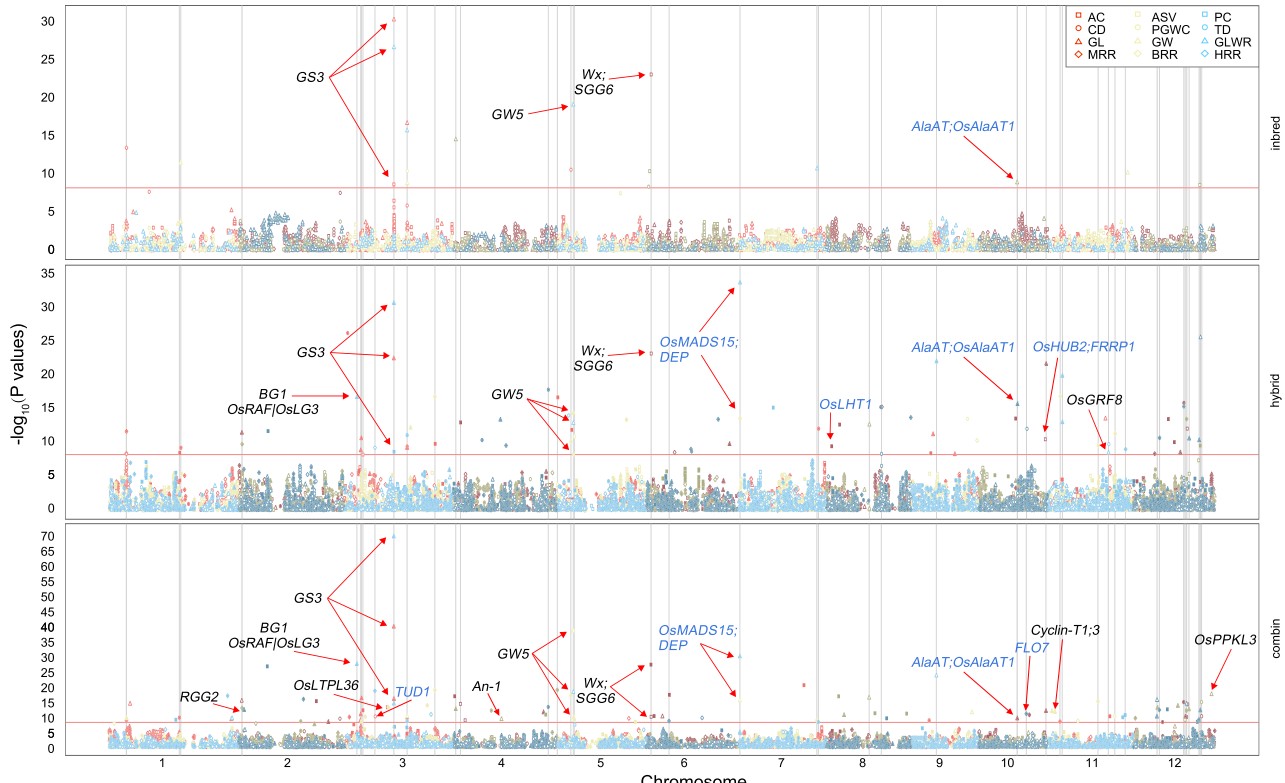

**Fig. 2 | Associations of additive and dominant effects analysed using the JPEG pipeline among inbred varieties, hybrids, and their combination.** The associations are illustrated as the negative −log10 P values of association tests on the 12 traits plotted against SNP positions for the 12 rice chromosomes. SNPs with additive and dominant effects are illustrated as open and filled symbols, respectively. The association analyses were conducted among inbred, hybrid, and their combined datasets. The inbred datasets include inbred observations and general combined ability. The hybrid datasets include hybrid observations and heterosis. The whole-genome association threshold was set to 1% after Bonferroni correction (the horizontal red lines). Known cloned genes are labelled at the nearest associated SNPs. Clone genes influencing the same quality trait are labelled in black. Genes reported to influence related quality traits are labelled in blue. An associated SNP is marked with a vertical line if it has significant effects in more than one analysis or is associated with more than one trait. Source data are provided as a Source data file.

Bayesian information and Linkage-disequilibrium Iteratively Nested Keyway (BLINK). When the phenotypes do not have a genetic effect (e.g., dominance), the corresponding genotype data has all zero elements.

We applied JPEG for gene mapping using three sources of data, i.e., inbred varieties, hybrid test crosses, and their combination. The combined analysis performed on the four datasets (V, T, G, and H) yielded higher power than analyses based on inbred varieties (V and G) or hybrid testcrosses (T and H). In total, 192 SNPs were identified as being associated with at least one of the 12 traits (Fig. 2 and Supplementary Figs. 7–9). Some of the associated SNPs had effects larger than one-half of one standard deviation (Supplementary Fig. 10). For example, the most significant SNP on AC is a known gene (Wx). Its genotypes had distinct phenotypic distributions in both inbred varieties and test crosses (Supplementary Fig. 11); this partially explained the bimodal phenotype distribution. After merging SNPs within 200 kb, 128 loci remained. The middle positions of the merged SNPs were used as the locations of the loci. The analyses with inbred parents, hybrids, and their combination identified 14, 68, and 89 loci, respectively. These loci were associated with at least one of the 12 traits in formats of additive or dominant effects. Most of the loci independently demonstrated additive or dominant effects, while 13 loci demonstrated both (Figs. 2 and 3 and Supplementary Data 1). There were 44 and 97 loci with additive and dominant effects, respectively. Among the 13 loci with both additive and dominant effects, eight were identified by all three analyses. Four of the eight loci were known genes, including GS3 for GL and GW5 for GW. The other two genes (Wx/ SSG6 and AlaAT|OsAlaAT1) are responsible for the AC. Among the 128

associated loci, 42 loci were located in or near (<200 kb) 17 known genes (Supplementary Data 1).

Many known genes were located within 200 kb of the identified loci. Three major genes of quality traits were identified: Wx (for AC), GS3 (for GL and GLWR), and GW5 (for GW and GLWR). Additionally, BG1 (for GW and GL) was found in the hybrid and combination datasets[24]. OsGRF8 was found in the hybrids for GLWR and in the combined dataset for AC. This gene is a growth-regulating factor that affects grain length, grain width, and grain starch granule size[25]. For the milling quality traits (MRR, BRR, and HRR), one significant locus was detected in the inbred dataset, compared with 19 in the hybrid dataset and 24 in the combined dataset. There were 17, 20, and 18 loci identified for GL, GW, and GLWR, respectively. Among the 18 loci identified for GLWR, a trait derived[26] from GL and GW, three were shared with GL, and six were shared with GW. The three loci that were shared by GL and GLWR were also shared with GW.

## Genetic effect/locus-based prediction

With the associated loci identified by GWAS, an immediate practical question was how to identify the new superior test crosses from the existing phenotypes and genotypes of inbred varieties and inferred genotypes of the hybrid test crosses. To answer this question, we first estimated the heritability of each trait to set a target for predictability. Heritabilities were estimated in inbred varieties for additive genetic effects only and in hybrid test crosses for additive, dominant, and total (additive + dominant) genetic effects (Fig. 4). The total heritability was considered as the target of accuracy to predict hybrid performance using genetic loci that could be manipulated through breeding.

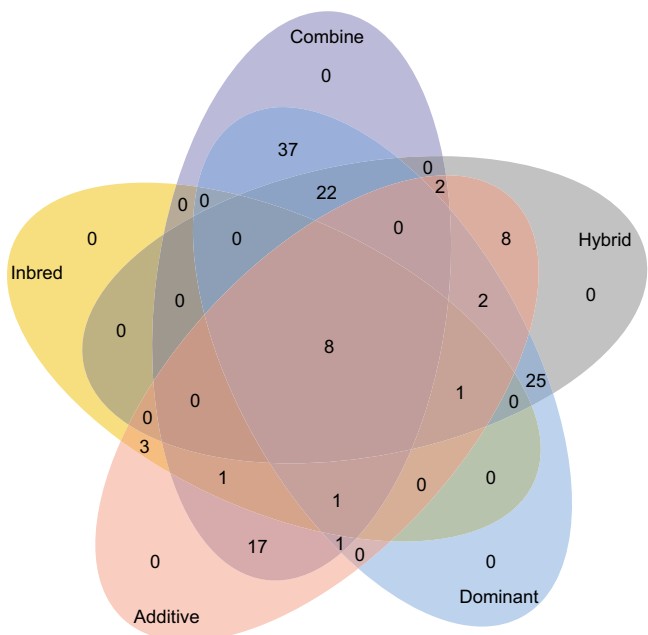

**Fig. 3 | Venn diagram of the numbers of associated loci identified by analyses with inbred varieties, hybrids, and their combination for additive and dominant genetic effects.** There were 128 loci associated with at least one of the 12 traits. Among these loci, 14, 68, and 89 were identified by analyses with inbred varieties, hybrids, and their combination, respectively. There are 9, 10, and 32 overlaps between inbred varieties and hybrids, between inbred varieties and the combination, and between hybrids and the combination, respectively. There were 44 and 97 loci with additive and dominant effects, respectively. There were 13 overlaps with both additive and dominant effects, including eight loci detected by all three analyses. Source data are provided as a Source data file.

As the 192 associated SNPs were identified by using the observations of the hybrids, the predictions made with these SNPs could not be used to assess applicability for future selection of superior testcrosses due to the overfitting problem. Therefore, we divided the entire population into training and testing populations and conducted GWAS on the training population only. The training population contained all of the inbred varieties. The hybrids were randomly divided into two groups. One group was taken as the testing population, and the other group was combined with inbred varieties as the training population. The newly detected SNPs were used to predict the phenotypes of the testing population. The observed phenotypes of the testing population were not used to determine the associated SNPs but instead were used to calculate the correlation coefficient with the predicted phenotypes. Cross validations were iterated until both groups were used as the testing population. This process was repeated 100 times. The mean of the squared correlation coefficients (R2_CV) for all iterations and replicates was used as the final assessment. The ratio of R2_CV to total heritability was used as the proportion of genetic variance explained by the detected SNPs.

Pyramiding the genetic loci for the 12 traits with additive and/or dominant effects in GS improved the accuracy of hybrid performance prediction, especially compared with the commonly used method, i.e., genomic best linear unbiased prediction (gBLUP). More than 45% of the total genetic variation was explained by the identified markers for all traits except one (HRR) with 30% of the variation explained. These results demonstrated that pyramiding genetically identified additive and dominant loci can provide substantial power to predict hybrid performance, presenting opportunities to identify superior crosses from existing phenotypes and genotypes of inbred varieties and hybrids (Fig. 4).

## Discussion

In hybrid rice production, seeds are commercial products, and their qualities are important to consumers. Hybrid plants are F1s, while the seeds they produce are F2s. The genotypes of F1 plants can be inferred from parent inbred varieties. The genotypes of F2s are segregating due to gamete recombination and chromosomal crossover[27]. Although hybrid seeds have different genotypes, their phenotypes are substantially determined by the mother plant F1s. Therefore, this study investigated the association between F1 genotypes and F1 phenotypes regarding the performance of the F2 seeds F1s produced.

Reducing false positives and increasing statistical power are both critical factors in GWAS[28]. Multiple loci models such as the Multiple Loci Mixed Model (MLMM), the Fixed And Random Model Circulating Probability Unification (FarmCPU), and BLINK have advantages over single-locus models in both reducing false positives and increasing statistical power[29]. In multiple loci models, markers are iteratively incorporated as covariates if they are determined to be associated with the trait during testing markers one at a time. This makes it feasible for multiple loci models to incorporate additive and dominant effects simultaneously by having both additive and dominant marker genotypes. Additive and dominant genetic effects can be realized by concatenating the additive genotypes and dominant genotypes side by side into one genotype matrix for testing. We chose BLINK over FarmCPU and MLMM because both FarmCPU and MLMM involve kinship relations derived from selected markers or all markers[30,31]. When additive and dominant marker genotypes are merged, the derived kinship loses its original property of being either additive or dominant kinship.

In this study, the GWAS with BLINK was used to analyse the genetic bases of 12 rice grain quality traits in 113 parental varieties and their 565 hybrid test crosses. A total of 192 significant SNPs clustered into 128 significant loci were associated with at least one of the 12 traits. Among these loci, 42 were located in or near (<200 kb) 17 known genes (Supplementary Data 1) with potential roles concerning rice grain quality. Several of the 128 associated loci were detected simultaneously in multiple analyses of the same trait or correlated traits. Some loci such as *Wx*[18], *GW5*[32], *GS3*[33], and *BG1*[34] were in or near the cloned quality genes.

In 2018, Liu et al.[35] reported that the rice grain yield quantitative trait locus *qLGY3* encodes the MADS-domain transcription factor *OsMADS1* that acts as a key downstream effector of G-protein βγ dimers. They demonstrated that combining the *OsMADS1* lgy3 allele with high-yield-associated dep1-1 and gs3 alleles represented an effective strategy for simultaneously improving both the productivity and end-use quality of rice. This study identified a locus near *OsMADS15*, a member of the *OsMADS1* gene family, that was significantly associated with two yield-related traits, GW and GLWR, in hybrid and combined datasets. *OsMADS15* may be another candidate gene that simultaneously affects grain yield and quality.

Although a substantial proportion (about one-third) of identified loci were known genes related to rice quality, this study was unable to provide insight into the relationship between additive and dominant loci, especially the occurrence of overdominant loci. Among the 128 identified loci, there were only 13 loci with both additive and dominant effects on one of the seven traits (Fig. 3). These traits were ASV, GL, GLWR, GW, MRR, PGWC, and TD (Supplementary Data 1). None of the traits had a locus with both additive and dominant effects. The results suggested that the dataset and the analyses did not have enough power to dissect the genetic architecture of overdominance on a particular trait.

In the current study, we limited our analyses to additive and dominant effects. JPEG can be extended to incorporate epistatic effects. For the additive genetic effects, genotypes are coded as 0 and 2 for the two homozygotes and 1 for the heterozygotes to form additive genotypes (A). For the dominant effects, genotypes are coded as 0

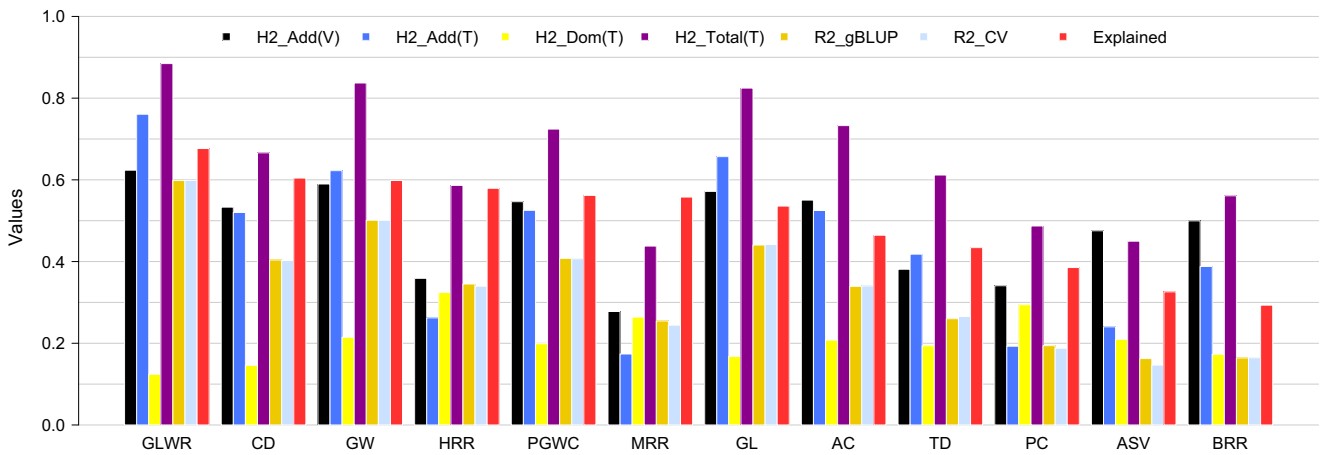

**Fig. 4 | Genomic heritability and predictability using identified additive and dominant loci.** The phenotypes of inbred varieties (V) were analysed to estimate additive heritability using additive kinship derived from the additive genotypes. The phenotypes of hybrid testcrosses (T) were analysed to estimate additive heritability and dominant heritability using additive kinship derived from the additive genotypes and dominant kinship derived from the dominant genotypes, respectively. The total heritability (additive + dominant) was calculated as the sum of the additive heritability and dominant heritability. The predictability of hybrids was evaluated through twofold cross-validation (CV). The reference population contained all parents and random sampled half from all offspring. The random sampling were repeated 100 times. The squared correlation coefficients (R2) between the observed and predicted values were used to examine predictability (R2_CV). Similarly, gBLUP was conducted and evaluated by squared correlation coefficients (R2_gBLUP). The ratio of R2_CV to the total heritability (H2_Total) indicated the percentages of the genome additive and dominant variances explained by the associated additive and dominant loci, which were over 50% for all traits except AC(46%), TD(43%), PC(38%) and ASV(32%). Source data are provided as a Source data file.

for homozygous and 1 for heterozygous to form dominant genotypes (D). A typical full model codes the haplotypes as dummy variables to represent the interaction between two loci[36]. Multiple reduced models can be built on the A and D matrices without specifying haplotypes, including additive by additive interaction (AA), dominant by dominant interaction (DD), and additive by dominant interaction (AD and DA) models. For example, in the DD model only an individual that is heterozygous for both loci is coded as 1, and the rest are coded as 0. In the AA model, an individual homozygous for the alternative alleles at both loci is coded as 4. An individual homozygous for the alternative alleles at one locus and heterozygous at the other locus is coded as 2. An individual heterozygous for both loci is coded as 1, and the rest are coded as 0. Even with the reduced models, the computation increases dramatically for the epistatic terms[37]. Restriction to loci with additive or dominant effects can make computation feasible at the cost of missing loci with epistatic effects.

Dominant genetic effects contribute to many complex human diseases and to phenotypic variation among animals and plants. Additive-nonadditive joint analysis is more powerful than additive effect analysis. When both parent inbred varieties and hybrid test crosses are available, general combined ability and heterosis can be derived to further assist in dissecting additive and dominant genetic effects. However, both methods and computing tools are lacking for the joint analysis of phenotypes, effects, and generations (JPEG). To overcome this problem, we developed a JPEG pipeline using BLINK software and analysed 113 male inbred varieties, five female inbred varieties, and their 565 (113×5) hybrid test crosses for 12 rice quality traits and 1,619,588 SNPs. Multiple loci were identified that not only included many known cloned genes but also demonstrated the feasibility of identifying superior crosses from parent inbred varieties and their partial hybrid test crosses.

## Methods
### Materials and field management
Following the diallel mating design, 115 indica rice accessions (Supplementary Data 2) as male parents were crossed with five male sterile varieties as female parents to produce hybrid test crosses. All seeds of the 115 male parental varieties were obtained from the China National Crop GenBank (https://www.cgris.net/cgris_english.html). The seeds included restorer varieties of 29 three-line wild-deficient hybrid rice and 86 accessions of microcore germplasm. In 2013, male parental varieties and hybrid test crosses were planted in Wuhan, China. The field trials were designed as randomized blocks and replicated twice, with 20 plants per field planted at a density of 5 × 8 inches. Field management, including irrigation, fertilizer application, and pest control, followed normal agricultural practices. The grains were harvested when fully ripe. In total, phenotypes of 12 rice quality traits were obtained from 113 parental varieties and 565 (113 × 5) hybrid test crosses. Two samples (*Dongqiubo* and *Guangqiuai*) with missing phenotypes were excluded from the analyses involving phenotypes.

### Phenotyping grain quality traits
After the materials had matured, three plants with uniform growth were selected from the middle eight plants, dried, and stored at room temperature for three months. The plants were then sent to the Agricultural College of Hunan Agricultural University for rice grain quality evaluation. A total of 12 rice quality traits of male parental varieties and hybrid test crosses were determined (Supplementary Data 3 and 4)[38], including the grain length (GL, mm); grain width (GW, mm); GL/GW ratio (GLWR); chalkiness degree (CD, %); percentage of grains with chalkiness (PGWC, %); transparent degree (TD, %); amylose content (AC %), alkali spreading value (ASV); PC (total protein content %); brown rice ratio (BRR, %); milled rice ratio (MRR, %) and whole milled rice rate (head rice ratio, HRR, %).

### Resequencing, genotyping, and imputation
For each parental variety, genomic DNA was prepared from a single plant for sequencing. A sequencing library was established following the Illumina protocol. The genomes of 118 parental varieties were sequenced on an Illumina HiSeq2500 platform with an 11× sequencing depth on average. Raw reads with an N ratio > 10%, those with more than 50% of the Q value (<5) ratio, and with average values < 15 were removed from the raw reads. Nipponbare was used as the reference genome (IRGSP-1.0, http://rapdb.dna.affrc.go.jp), and BWA software[39] was used for all paired-end read mapping using default parameters. PCR duplicates were removed by "rmdup" in SAMtools version 1.12.

SNP calling was performed using the GATK UnifiedGenotyper set for diploids using default filtering settings. SNP quality control was conducted by deleting SNPs with a missing rate >20% and minor allele frequency <5%. In total, 1,619,588 SNPs were obtained (Supplementary Data 5). Missing genotypes were imputed by NPUTE[40] (version 4.0). Hybrid test cross genotypes were inferred using parental SNP genotypic information. The genotypes were coded for both additive and dominant genotypes. The additive genotypes were coded as 0 and 2 for the two homozygotes and 1 for the heterozygotes. The homozygote with a nucleotide in lower alphabetical order was assigned the numeric value of 0, and the other homozygote with a nucleotide in higher alphabetical order was assigned the numerical value of 2. The dominant genotypes were coded as 0 for the two homozygotes and 1 for the heterozygotes.

### Phylogenetic and population structures

Neighbour-joining (NJ) trees and principal component analysis (PCA) plots were used to infer the structures of the 118 rice parental varieties and 565 hybrid test crosses. A pairwise distance matrix derived from the simple matching distance for all the SNP sites was calculated to construct unweighted NJ trees using the software SNPhylo[41] (version 20140116), and phylogenetic trees were drawn by iTOL online (http://itol.embl.de/). The program Admixture (version 1.3) was used to perform optimization of the number of subpopulations (K) in the range 2 to 10[42]. PCA was conducted using GAPIT[43] (version 3.0) and was performed separately for parental varieties and hybrid test crosses. Genome-wide linkage disequilibrium (LD) was estimated by PLINK[44] (version 1.9) as pairwise $r^2$ values among SNPs within a window containing no more than 99999 SNPs. The parameters for the commands were "--r2 --ld-window 99999 --ld-window-r2 0". The first "r2" specifies LD as $r^2$. The second "r2" followed by "0" specifies the output for $r^2$ values of 0 or above, which is everything. The parameter of "99999" specifies the windows size defined as the number of SNPs.

### Phenotypic and heterotic statistics

The phenotypes of a trait were denoted V for the inbred varieties and T for the test cross. Two additional types of observations were derived from V and T, i.e., general combining ability (G) (Supplementary Data 6) for inbred varieties and heterosis (H) (Supplementary Data 7) for hybrid testcrosses.

The general combined ability of an inbred variety i was derived from its test cross using the following formula:

$$G_i = \bar{y}_i - \bar{y}_{..} \tag{1}$$

where $G_i$ represents the G of the ith paternal parent; $\bar{y}_i$ represents the mean phenotypic value of test crosses derived from the ith parent, and $\bar{y}_{..}$ represents the mean phenotypic value of all the hybrid test crosses.

Heterosis was defined as the difference between a testcross and the average of its parents as indicated by the following formula:

$$H_{ij} = T_{ij} - (V_i + V_j)/2 \tag{2}$$

where $T_{ij}$ represents the phenotypic value of the test cross hybrid derived from the ith male parent and jth female parent; $V_i$ represents the phenotype of the ith male parental varieties, and $V_j$ represents the jth female parent phenotype.

### GWAS pipeline across parents and hybrids for both additive and dominant effects

The four types of observations (V, T, G, and H) were normalized within each type to eliminate the differences between scales and averages. Observations were vertically concatenated as a single phenotype vector (Y) in the combined analysis, and the corresponding genotype matrix was generated. V and G shared the same genotype matrix, while

T and H shared the same genotype matrix (Fig. 1). The additive genotype matrix and dominant genotype matrix were horizontally concatenated (left and right) for GWAS with the BLINK[29] multiple loci model implemented in GAPIT[43] (version 3.0). We named this pipeline the joint analysis of phenotypes, effects, and generations (JPEG). This JPEG pipeline contains extraction and a combination of phenotypes, GWAS, and Genomic Prediction. The source code is available at GitHub (https://github.com/jiabowang/JPEG).

Homozygous genotypes are coded as 0 and 2, while heterozygous genotypes are coded as 1 in the additive genotype matrix (A). Similarly, both homozygous genotypes are coded as 0, while the heterozygous genotype is coded as 1 in the dominant genotype matrix (D). The dominant genotypes are all 0 s for V and G. An association test is based on whether an SNP has a non-zero additive effect or a non-zero dominant effect. An SNP with a non-zero additive effect only is defined as an additive locus. An SNP with a non-zero dominant effect only is defined as a dominant locus. An SNP with both a non-zero additive effect and a non-zero dominant effect is defined as an additive and dominant locus.

Association tests on additive effects were conducted by using the general linear model listed in formula (3) and on dominant effects using the general linear model in formula (4).

$$Y_i = b_0 + B_{i1}b_1 + B_{i2}b_2 + \ldots + B_{it}b_t + A_{ij}a_j + e_i \tag{3}$$

$$Y_i = b_0 + B_{i1}b_1 + B_{i2}b_2 + \ldots + B_{it}b_t + D_{ij}d_j + e_i \tag{4}$$

Here, $Y_i$ is ith observation; $b_0$ is the overall population mean; $B_{i1}$, $B_{i2}$, …, and $B_{it}$ are the covariates for the ith observation, including the first three principal components, dummy variables of female parents for the testcross, and selected additive and dominant SNPs; $b_1, b_2, …, b_t$ are the corresponding fixed effects of the covariates; $A_{ij}$ is the additive genotype of the ith individual on the jth SNP; $a_j$ is the corresponding fixed additive effect of the jth SNP; $D_{ij}$ is the dominant genotype of the ith individual on the jth SNP; $d_j$ is the corresponding fixed dominant effect of the jth SNP; and $e_i$ is the random residual having a distribution with a mean of zero and a variance of $\sigma_e^2$.

Association tests were first conducted for each of the marker on additive effects and afterward for dominant effects. At the end of the iterations for testing additive and dominant marker effects, the associated marker effects, either additive or dominant, were selected as covariates in the model. The process was iterated until no further SNPs could be selected as covariates. The association was determined with the threshold of 1% type I error after Bonferroni multiple test correction on both additive and dominant markers.

### Heritability estimation among inbred varieties and hybrids

Additive genotype matrices were used to derive the additive genetic kinship matrices among inbred varieties and hybrid test crosses. Similarly, a dominant genetic marker matrix was used to derive dominant kinship relations among hybrid test crosses. Additive kinship and dominant kinship matrices define the covariance structure of the additive genetic effect and the dominant genetic effect. The kinship matrices were calculated with the VanRaden algorithm[45]. A mixed linear model with random additive and dominant genetic effects was solved by the BGLR[46] software package using the Gaussian processes (RKHS) algorithm to estimate the additive genetic variance, dominance genetic variance, and residual variance. The proportions of the additive genetic variance, dominance genetic variance, and total genetic variance (additive + dominant) over the total variance (residual variance included) were calculated as the corresponding additive heritability, dominant heritability, and total heritability, respectively.

## Genomic best linear unbiased prediction

We conducted cross-validations using the classical genomic Best Unbiased Linear Prediction (gBLUP) as the reference to evaluate the capability to select superior test crosses. All the parent inbred varieties were used as a training population. The hybrids were randomly divided into two equal groups. One group was selected as a testing population, and the other group was joined with the parent inbred varieties as the training population. In other words, all $V$ and $G$ belonged to the training group, while half of $T$ and $H$ belonged to the training group and the other half belonged to the testing group. Additive kinship was calculated from the additive genotypes corresponding to $V$, $T$, $G$, and $H$ using the VanRaden algorithm[45] implemented in GAPIT[43]. The observations corresponding to testing population were set to "NA" to run gBLUP in GAPIT for prediction. The correlations between observed and predicted observations were calculated for the hybrids in the testing population. The testing of hybrids was iterated until both groups were tested. All processes were repeated 100 times. The mean squared correlation coefficient across all groups and replicates was used to assess the accuracy of the gBLUP.

## Genetic effect-locus-based prediction

The most critical goal in hybrid breeding is to identify new superior crosses with existing genotypes of inbred varieties and phenotypes of inbred varieties and their hybrids. The predictions for hybrids using the associated additive and dominant loci identified previously in the GWAS could not be used to evaluate their predictability due to potential model overfitting because these loci were derived from the hybrids. To assess the capability, we repeated the GWAS on the training population only during the cross-validation analyses. Similar to the cross-validation for the gBLUP, we randomly divided the hybrids into two equal groups. One group of hybrids was selected as a testing population. The remaining hybrid group and the parent inbred varieties were used as the training population. The GWAS model with both additive and dominant genetic effects was implemented using BLINK on the training population only. The newly-detected SNPs with additive effects and SNPs with dominant effects in the final iteration of BLINK were used as explanatory variables in GAPIT to predict the phenotypes of the hybrids in the testing population using the gBLUP model. The kinship was derived from both additive and dominant genotypes. The correlation coefficients between observed and predicted observations were calculated for the hybrids in the testing population. The testing population was iterated until both groups were tested. All processes were repeated 100 times. The mean squared correlation coefficient across all groups and replicates was used to assess the capability to identify new superior crosses.

## Reporting summary

Further information on research design is available in the Nature Portfolio Reporting Summary linked to this article.

## Data availability

The raw DNA sequencing data of 120 rice inbred lines analysed in the current study are available in the NCBI Sequence Read Archive under accession numbers SRP080763 and SRP080834. Source data are provided with this paper.

## Code availability

JPEG source code is available at GitHub [https://github.com/jiabowang/JPEG].

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

## Acknowledgements

This work was partially supported by the National Key Technology Research and Development Program (2016YFD0100101, Z.H.), the Open Research Fund of State Key Laboratory of Hybrid Rice (Hunan Hybrid Rice Research Center (2019KF05, L.L.), Wuhan University (KF201912, L.L.), the Natural Science Foundation of Hunan Province (2020JJ4039, L.L.), Special Funds for Construction of Innovative Provinces in Hunan Province (2021NK1011, Z.Y.), the key Research and Development Program of Hubei Province (2021BBA223, X. Zheng), the Sichuan Science and Technology Program, China (Award #s 2021YJ0269 and 2021YJ0266, J.W.), the USDA National Institute of Food and Agriculture (Hatch project 1014919 and Award #2020-67021-32460, Z.Z.) and the Washington Grain Commission (Award # 126593 and 134574, Z.Z.). The authors thank Prof. Sibin Yu and Tongmin Mou from Huazhong Agricultural University, Prof. Shuangcheng Li from Sichuan Agricultural University, Prof. Shuzhu Tang from Yangzhou University, Prof. Guanghua He from Southwest University, and Prof. Huaxiong Qi from Hubei Academy of Agricultural Sciences for providing materials.

## Author contributions

L.L., X. Zheng, and J.W. performed data analyses and drafted the manuscript. X. Zhang, X.H., L.X., S.S., J.S., W.T., and Z.Y. assisted in data analysis and the discussion. Y.D. and X. Zheng developed the populations. X. Zheng and X.H. assisted in field data collection. Z.H. conceived the project and planned and secured extramural funds. Z.Z. designed the data analyses and revised the manuscript.

## Competing interests

The authors declare no competing interests.
