## [Peer Review File · Nature Communications]

Additive and Dominant Loci Jointly Pyramiding the Grain Quality of Hybrid RiceReviewers' Comments:

Reviewer #1:

Remarks to the Author:

The work is neither novel nor of high quality. I do not think the paper is suitable for publication in Nature Communications.

Firstly, genome-wide associations for rice grain quality traits have been reported in many previous papers. Moreover, genomic resources of diverse parental lines of hybrid rice are also reported in many papers.

Secondly, all of the main figures, except figure 2, are very simple, lacking useful insights for rice genetics or take home messages for hybrid rice breeding.

Lastly, the authors "guessed" several candidate genes, as indicated in figure 2. However, no functional validations were performed in this work. Follow-up functional experiments, at least for 1-2 newly identified genes, are obviously required for high-quality journals.

Reviewer #2:

Remarks to the Author:

In this study, the authors used 113 rice inbred lines as male parents and 5 rice inbred lines as female parents (or tester lines) to make 565 F1 hybrids. The parental lines and their 565 F1 hybrids were evaluated on major grain quality traits, and all parental lines were genotyped by the whole-genome sequencing technology. A joint analysis method called JPEG was developed to analyze the compound data sets. By using the JPEG method, a large amount of genetic loci were identified, having both additive and dominant effects on the evaluated quality traits. Results and methodologies present in this study are of importance in rice genetics and hybrid breeding. Some specific comments are given below.

1. The study reported 44 loci of additive effects, 97 loci of dominant effects, and 13 of them had both additive and dominant effects. Over-dominance is also commonly found in heterotic studies, especially in maize. It would be useful to tell whether over-dominance is present for those quality traits.
2. Line 35: what does 30% stand for? Does it mean 'all traits except one'?
3. There are many abbreviations throughout the text, some are common, but some are less common. It would be better if the authors give a list of abbreviations for the convenience of less-professional readers.
4. Line 90: would 'realize' be better than 'recognize'?
5. Lines 96-98: The manuscript is generally well-written. But there is still a space of improvement. This sentence is hard to understand. In addition, if a and d are the additive and dominant effects at one locus, $d=a$ can be called dominant; $d=-a$ can be called recessive; $d=0$ can be called additive; over dominance and over recessive can also be defined from the relative size of a and d . What is the essential difference between the dominant and recessive models?
6. Line 100: Is 'joint additive and dominant effect model' more clear?
7. Line 272: To say 'genotypes remain dynamic' is not logic. It is more common to say 'genotypes are segregating or heterogeneous'.
8. Line 346: Here, diallel mating design is more relevant. In NCII, parents are randomly drawn from a random-mating population, which are heterozygous in genotypes. Parents used in this study are homozygous.
9. Lines 402 and 403: the subscript l is a typo, which should be i .
10. Lines 409-412: as mathematical symbols, H , T , and V should have the italic format.
11. Figure 1: This figure is key to the JPEG method. I wish it could be more indicative and meaningful. The meaning of solid and dash arrows is not clear to me. The meaning of Bb , Aa , and Dd is not clear

either.

Reviewer #3:

Remarks to the Author:

The authors developed 565 hybrids of rice from the crosses of 113 male lines and 5 female lines. From these hybrids, they conducted association studies for 12 grain quality traits. In total, they identified 128 loci, including many known quality related genes. Through cross-validation, they also demonstrated the feasibility of identifying superior test crosses from parental inbred varieties and their partial hybrid test crosses. As similar types of data exist everywhere, this study has a potential impact on enhancing discoveries through advanced data analyses. The manuscript is well organized and well written. With some degree of revision, it can be accepted in this journal.

Major comments

1. It is unclear whether the JPEG pipeline also includes genomic prediction or not beyond GWAS. The manuscript lacks detailed information about data analysis and it may be hard for readers to replicate the analyses.
2. The source code of JPEG pipeline should be available to the public.
3. The population structure analysis suggested five sub-populations (with minimum error (Figure S5)). However, the authors only used the first three PCs (Line 428) in the GWAS analysis. Some justification is required.
4. The authors should make statements carefully to avoid self-conflict. For example, the authors claimed that (Page 3, line 45) "rice quality traits are typically correlated with yield related traits negatively". However, the authors also demonstrated that some rice quality traits are positively correlated with yield related traits.
5. The English of the manuscript needs substantial editing. It is better to ask a native English writer for help.

Minor comments

1. Figure S4 (B): Suggest displaying X-axis at the maximum of 500 kb. With the current form, half of the graph has no information.
2. Figure S6 (E and F). Every female line has about the same number of outliers (4). This very strange phenomenon needs to be explained in the text.
3. Figure 1: The model does not include covariates outside of markers such as PCs and dummy variables which are described in the main text.

Reviewer #4:

Remarks to the Author:

The manuscript reports a pipeline for jointly analyzing inbred varieties and their hybrids in the context of a genome-wide association study. The proposed pipeline is specifically applied to grain quality in rice. The authors generated 565 hybrids by crossing 113 male inbred parents with 5 female inbred parents. They phenotyped both parents and hybrids for 12 quality traits and genotyped the parents with whole-genome sequencing. Sequencing yielded 1,619,588 SNPs and genotype in hybrids was

inferred from parental genotypes. The authors derived general combining abilities (G) and heterosis (H) from the phenotypic data on parents (V) and hybrids (T) and developed a pipeline to jointly analyze the phenotypic data (G, H, V, and T) in a genome-wide association study. The pipeline relies on the BLINK model, and consists in staging the data together, and using the same genotypic data for T and H, and for V and G, respectively. They also performed separate models on hybrids and parents, using both phenotypic data and derived phenotypes. They detected more loci with the joint model, combining hybrids and parents, than with the separate ones. Some of the detected loci were close to known genes which appeared to be relevant for the phenotypes under study. Finally, they made use of the proposed pipeline to predict hybrid phenotypes within a cross-validation scheme. Interestingly, the prediction explained a fairly large part of the genomic heritability.

Overall I found this manuscript interesting. Nevertheless, I have several concerns/suggestions that I would like the authors to address to improve the quality of their manuscript.

- From the description, it seems to me that the proposed pipeline turns out to be a multivariate linear model. Could you please provide more details on the model, for instance by writing the equation?
- If the model is effectively a multivariate linear model, then I believe that the covariance between phenotypes should be modeled. In the current version, is the covariance between the 4 phenotypes (V, T, G H) taken into account?
- Could you please provide QQ-plots for the p-values of the association tests in both separate and combined models?
- Could you please provide, as a supplementary figure, boxplots to illustrate the effect of each detected SNP on the corresponding phenotype (V, T, G, H for the combined model; V, G for the Inbred model; T, H for the Hybrid model)?
- From the Venn diagram (Figure 3), it is apparent that the combined model detects 37 loci that were not detected by separate models. It is also apparent that separate models detect loci not detected by the combined model (36 for the Hybrid model and 3 for the Inbred model). Could you please comment on this?
- The prediction is interesting and well performed to avoid overfitting. The ratio of R^2_{CV} to the total genomic heritability is informative. Yet, to be completely convincing I would suggest running classical genomic prediction models (ridge regression and lasso) using the same cross-validation settings as a benchmark for the proposed pipeline, and to compare the resulting predictability with those obtained with the proposed pipeline. Also one can wonder, what is the predictability of loci identified with separate models, and in turn how much predictability is gained from using the combined model?
- The distribution of several phenotypes is bimodal as mentioned in the text (I134). Is the bimodality due to the large effect of some loci? As already mentioned, boxplots illustrating the effects of SNPs would help interpret some results.
- The admixture analysis is almost not commented on in the text, apart from the optimal number of clusters. Maybe it is not useful. I would suggest either commenting more on this analysis or dropping it from the manuscript.
- The discussion section lacks references to the literature. In the current version, there is only one citation in the entire section (Liu et al. ref 26, I303). This is quite unusual because the authors are supposed to discuss their results in light of the literature in this particular section. Please add citations to support your statements.
- In the method section, some crucial details are lacking concerning the detection and typing of SNPs. Please provide all tools and parameters used to ensure the reproducibility of your research.
- For LD computation, could you please add what does the plink command (I392-393) mean?
- In equation 1 (I402): what does the subscript "l" stand for? Could you also please make sure that the subscript match with those used in equation (2).
- For the heritability estimation, could you please give the method used to compute the additive and dominant genomic relationship matrices?
- I120 "combinability" should be "combining ability".
- I281 "modes" should be "models".

REVIEWER COMMENTS

Reviewer #1 (Remarks to the Author):

The work is neither novel nor of high quality. I do not think the paper is suitable for publication in Nature Communications.

Response: Thank you for your prompt evaluation of being the #1 reviewer of our manuscript. We appreciate your time and feel sorry that our manuscript did not communicate well with the journal we have been working with. Based on the editor's suggestions and the comments from the other three reviewers, we revised our manuscript accordingly to improve its readability.

Firstly, genome-wide associations for rice grain quality traits have been reported in many previous papers. Moreover, genomic resources of diverse parental lines of hybrid rice are also reported in many papers.

Response: We agree with your judgment that GWAS on rice quality traits have been intensively investigated, and diversified genetic resources are widely available, which motivated our study. The other reviewers shared the same view. For example, Reviewer #2 said: "Results and methodologies present in this study are of important in rice genetics and hybrid breeding". Reviewer #3 said: "As similar types of data exist everywhere, this study has a potential impact on enhancing discoveries through advanced data analyses". We feel sorry we did not make our objectives clear enough for everyone in our efforts to develop a new analytical pipeline so that the existing and future data can be analyzed differently with greater success. We revised our manuscript to clarify our objectives.

Secondly, all of the main figures, except figure 2, are very simple, lacking useful insights for rice genetics or take-home messages for hybrid rice breeding.

Response: We agree with you and other reviewers as well that the main figures need improvement for clarity. As reviewer #2 suggested specifically, Figure 1 needs more definitions of terms such as Bb, Aa, and Dd. We revised the figure and others as well for clarity and provided the biological interpretation in the main text from the aspects of rice genetics and hybrid rice breeding.

Lastly, the authors "guessed" several candidate genes, as indicated in figure 2. However, no functional validations were performed in this work. Follow-up functional experiments, at least for 1-2 newly identified genes, are obviously required for high-quality journals.

Response: We agree that functional validation of the candidate genes is critical for our ultimate understanding and applications in the rice community. This was also the reason for inspiring us not only to provide a new pipeline to analyze existing and future data with greater success but also to provide the analytical results with many genetic loci that have not been reported before for controlling rice quality traits. The rice community will be benefited more from seeing these loci now than seeing the cloning of 1-2 genes a few years later. We revised our manuscript to clarify the argument. Again thank you for your thoughts and your time in voluntarily evaluating our manuscript.

Reviewer #2 (Remarks to the Author):

In this study, the authors used 113 rice inbred lines as male parents and 5 rice inbred lines as female parents (or tester lines) to make 565 F1 hybrids. The parental lines and their 565 F1 hybrids were evaluated on major grain quality traits, and all parental lines were genotyped by the whole-genome sequencing technology. A joint analysis method called JPEG was developed to analyze the compound data sets. By using the JPEG method, a large amount of genetic loci were identified, having both additive and dominant effects on the evaluated quality traits. Results and methodologies present in this study are of importance in rice genetics and hybrid breeding. Some specific comments are given below.

Response: Thank you for briefing our experiment, analyses, and findings and your comments on the important roles of our results and methodologies in rice genetics and hybrid breeding.

1. The study reported 44 loci of additive effects, 97 loci of dominant effects, and 13 of them had both additive and dominant effects. Over-dominance is also commonly found in heterotic studies, especially in maize. It would be useful to tell whether over-dominance is present for those quality traits.

Response: We agree this is important scientific question. Our experiment had the statistical power to identify individual loci with additive or dominant genetic effects on one of the twelve rice quality traits in one of the three analyses. However, the probability is low to have sufficient statistical power to detect both additive and dominant genetic effects for the same traits and the same analyses. Especially, the analyses with inbred is not able to estimate the dominant effects. Answering your important question is beyond the capability of our experiment.

2. Line 35: what does 30% stand for? Does it mean 'all traits except one'?

Response: You are correct. We revised the sentence for clarity.

3. There are many abbreviations throughout the text, some are common, but some are less common. It would be better if the authors give a list of abbreviations for the convenience of less-professional readers.

Response: We summarized all the abbreviations in supplementary (Box 2).

4. Line 90: would 'realize' be better than 'recognize'?

Response: We accepted your suggestion.

5. Lines 96-98: The manuscript is generally well-written. But there is still a space of improvement. This sentence is hard to understand. In addition, if a and d are the additive and dominant effects at one locus, $d=a$ can be called dominant; $d=-a$ can be called recessive; $d=0$ can be called additive; over dominance and over recessive can also be defined from the relative size of a and d . What is the essential difference between the dominant and recessive models?

Response: We appreciate your overall comment on the writing, and we have gone through the entire manuscript to further improve the readability. Dominant and receive are relative

to each other and dependent on genotypes. We saw the problem of generalizing the difference between dominant and recessive models. Therefore, we removed the statement, which is irrelevant to the major findings.

6. Line 100: Is 'joint additive and dominant effect model' more clear?

Response: Yes, we revised the sentence accordingly.

7. Line 272: To say 'genotypes remain dynamic' is not logic. It is more common to say 'genotypes are segregating or heterogeneous'.

Response: Yes, we revised the sentence accordingly.

8. Line 346: Here, diallel mating design is more relevant. In NCII, parents are randomly drawn from a random-mating population, which are heterozygous in genotypes. Parents used in this study are homozygous.

Response: We revised the sentence accordingly.

9. Lines 402 and 403: the subscript *l* is a typo, which should be *i*.

Response: We fixed the typo.

10. Lines 409-412: as mathematical symbols, *H*, *T*, and *V* should have the italic format.

Response: We placed them in italic.

11. Figure 1: This figure is key to the JPEG method. I wish it could be more indicative and meaningful. The meaning of solid and dash arrows is not clear to me. The meaning of *Bb*, *Aa*, and *Dd* is not clear either.

Response: We appreciate that you consider the figure as the key to illustrate the method. We revised the figure and detailed the description of the items, such as the lines and notations.

Reviewer #3 (Remarks to the Author):

The authors developed 565 hybrids of rice from the crosses of 113 male lines and 5 female lines. From these hybrids, they conducted association studies for 12 grain quality traits. In total, they identified 128 loci, including many known quality related genes. Through cross-validation, they also demonstrated the feasibility of identifying superior test crosses from parental inbred varieties and their partial hybrid test crosses. As similar types of data exist everywhere, this study has a potential impact on enhancing discoveries through advanced data analyses. The manuscript is well organized and well written. With some degree of revision, it can be accepted in this journal.

Response: Thank you for briefing our experiment, analyses and findings and your comments on the potential impact of our study and writing of the manuscript.

Major comments

1. It is unclear whether the JPEG pipeline also includes genomic prediction or not beyond

GWAS. The manuscript lacks detailed information about data analysis and it may be hard for readers to replicate the analyses.

Response: The JPEG pipeline includes both GWAS and genomic prediction. We added details in the data analyses for readers to replicate our study or conduct similar analyses on their data.

2. The source code of JPEG pipeline should be available to the public.

Response: The source code of JPEG pipeline is available at GitHub (<https://github.com/jiabowang/JPEG>).

3. The population structure analysis suggested five sub-populations (with minimum error (Figure S5)). However, the authors only used the first three PCs (Line 428) in the GWAS analysis. Some justification is required.

Response: This is a very interesting concern. As a single covariate can capture multiple levels of the input variable, there is no direct connection between the number of sub-populations and the number of PCs as covariates to model population structure.

4. The authors should make statements carefully to avoid self-conflict. For example, the authors claimed that (Page 3, line 45) “rice quality traits are typically correlated with yield related traits negatively”. However, the authors also demonstrated that some rice quality traits are positively correlated with yield related traits.

Response: We revised the sentence indicate their correlation but not direction.

5. The English of the manuscript needs substantial editing. It is better to ask a native English writer for help.

Response: We polished the manuscript with help from native speakers.

Minor comments

1. Figure S4 (B): Suggest displaying X-axis at the maximum of 500 kb. With the current form, half of the graph has no information.

Response: We revised the figure with the maximum of 500 kb to make the figure more informative.

2. Figure S6 (E and F). Every female line has about the same number of outliers (4). This very strange phenomenon needs to be explained in the text.

Response: We added the description to explain the phenomenon. They are four male parent outliers.

3. Figure 1: The model does not include covariates outside of markers such as PCs and dummy variables which are described in the main text.

Response: The covariates can be initiated with explanation variables such as PCs. We revised our manuscript to reflect the initiation.

Reviewer #4 (Remarks to the Author):

The manuscript reports a pipeline for jointly analyzing inbred varieties and their hybrids in the context of a genome-wide association study. The proposed pipeline is specifically applied to grain quality in rice. The authors generated 565 hybrids by crossing 113 male inbred parents with 5 female inbred parents. They phenotyped both parents and hybrids for 12 quality traits and genotyped the parents with whole-genome sequencing. Sequencing yielded 1,619,588 SNPs and genotype in hybrids was inferred from parental genotypes. The authors derived general combining abilities (G) and heterosis (H) from the phenotypic data on parents (V) and hybrids (T) and developed a pipeline to jointly analyze the phenotypic data (G, H, V, and T) in a genome-wide association study. The pipeline relies on the BLINK model, and consists in staging the data together, and using the same genotypic data for T and H, and for V and G, respectively. They also performed separate models on hybrids and parents, using both phenotypic data and derived phenotypes. They detected more loci with the joint model, combining hybrids and parents, than with the separate ones. Some of the detected loci were close to known genes which appeared to be relevant for the phenotypes under study. Finally, they made use of the proposed pipeline to predict hybrid phenotypes within a cross-validation scheme. Interestingly, the prediction explained a fairly large part of the genomic heritability.

Overall I found this manuscript interesting. Nevertheless, I have several concerns/suggestions that I would like the authors to address to improve the quality of their manuscript.

Response: Thank you for briefing our experiment, analyses, and findings. We appreciate that you found our manuscript interesting.

- From the description, it seems to me that the proposed pipeline turns out to be a multivariate linear model. Could you please provide more details on the model, for instance by writing the equation?

Response: Yes, you are correct. A multivariate linear model could be another way to interpret our data which is complex with multiple variables such as original observation and derived general combination ability and heterosis on objects (inbreds and hybrids) with complete missing values on some of the dependent variables. Fortunately, when the objects are decomposed into their genetic bases (SNPs), the model becomes much simpler, and missing data is dramatically reduced. For example, the additive SNP effects exist for both inbreds and hybrids and can be modeled as fixed effects.

- If the model is effectively a multivariate linear model, then I believe that the covariance between phenotypes should be modeled. In the current version, is the covariance between the 4 phenotypes (V, T, G H) taken into account?

Response: You are correct that we should model and consider the covariance between the four phenotypes if we choose a random model, which is much more complex. As described above, we used fixed SNP effects as explanation variables. For the residual random effects, we assume null covariance for the dependent variables.

- Could you please provide QQ-plots for the p-values of the association tests in both separate and combined models?

Response: We added the plot as requested.

- Could you please provide, as a supplementary figure, boxplots to illustrate the effect of each detected SNP on the corresponding phenotype (V, T, G, H for the combined model; V, G for the Inbred model; T, H for the Hybrid model)?

Response: We added a plot to display the effects (Figure S7).

- From the Venn diagram (Figure 3), it is apparent that the combined model detects 37 loci that were not detected by separate models. It is also apparent that separate models detect loci not detected by the combined model (36 for the Hybrid model and 3 for the Inbred model). Could you please comment on this?

Response: Inference of an association between a phenotype and a marker is a combination of the phenotypic differences among the marker genotypes, the error against with, and the threshold of multiple test correction. The combined analysis has the advantage of a larger degree of freedom for the error term and the disadvantage of a heavier penalty on multiple test corrections than the separated analyses, such as the inbred model and the hybrid model. These differences led to the phenomenon even when the differences among the marker genotypes stay the same, which is rare due to random sampling.

- The prediction is interesting and well performed to avoid overfitting. The ratio of R^2_{CV} to the total genomic heritability is informative. Yet, to be completely convincing I would suggest running classical genomic prediction models (ridge regression and lasso) using the same cross-validation settings as a benchmark for the proposed pipeline, and to compare the resulting predictability with those obtained with the proposed pipeline. Also one can wonder, what is the predictability of loci identified with separate models, and in turn how much predictability is gained from using the combined model?

Response: As you requested, we added the performance of gBLUP as a benchmark (please see updated Figure 4). gBLUP is identical ridge regression mathematically proved by Dr. VanRaden (Journal of Dairy Science, 91 (11), 4414-4423). GBLUP was performed with EMMREML package. The results suggested that our proposed method was superior to gBLUP.

- The distribution of several phenotypes is bimodal as mentioned in the text (l134). Is the bimodality due to the large effect of some loci? As already mentioned, boxplots illustrating the effects of SNPs would help interpret some results.

Response: You are correct. We added a figure in the supplementary (Figure S8) to demonstrate phenotype distributions within genotypes of the most associated markers for all traits.

- The admixture analysis is almost not commented on in the text, apart from the optimal

number of clusters. Maybe it is not useful. I would suggest either commenting more on this analysis or dropping it from the manuscript.

Response: We added more interpretation to the analysis.

- The discussion section lacks references to the literature. In the current version, there is only one citation in the entire section (Liu et al. ref 26, l303). This is quite unusual because the authors are supposed to discuss their results in light of the literature in this particular section. Please add citations to support your statements.

Response: We added more citations.

- In the method section, some crucial details are lacking concerning the detection and typing of SNPs. Please provide all tools and parameters used to ensure the reproducibility of your research.

Response: We described the tools and parameters in the *Material and Methods* section.

- For LD computation, could you please add what does the plink command (l392-393) mean?

Response: We described the parameters.

- In equation 1 (l402): what does the subscript "l" stand for? Could you also please make sure that the subscript match with those used in equation (2).

Response: We corrected the typo.

- For the heritability estimation, could you please give the method used to compute the additive and dominant genomic relationship matrices?

Response: The additive and dominant kinship was calculated from the additive genotypes and dominant genotypes, respectively, by using the VanRaden algorithm (Journal of Dairy Science, 91 (11), 4414-4423). The estimation of the variance component was conducted by the "RKHS" model implemented in the BGLR package. We added the details in the revised manuscript.

- l120 "combinability" should be "combining ability".

Response: We accepted your suggestion and revised the manuscript.

- l281 "modes" should be "models".

Response: We corrected the typo

Reviewers' Comments:

Reviewer #1:

Remarks to the Author:

The manuscript is not improved. Only Figure 2 provides useful results with several QTLs and candidate genes identified. The contents of all the other figures are far from the standard of Nature Communications.

Reviewer #2:

Remarks to the Author:

I think the authors made a significant improvement in this revision. Most comments from this reviewer have been handled properly. I have only a few minor comments to this revision.

(1) In abstract, the authors stated that "There were 44 and 97 loci with additive and dominant effects, respectively, including 13 overlaps." One locus can have both additive (a) and dominant (d) effects. Degree of dominance, i.e. d/a , can take any value. It is not clear how the detected loci were classified to be additive or dominant.

(2) In genetic studies on heterosis in maize or rice, over-dominance was frequently observed. If no over-dominance was observed in this study, the authors may want to discuss the possible reasons.

(3) I understand that GLWR (i.e. GL/GW) is an important quality index in rice, but the use of a ratio of two measured traits in genetic analysis may be problematic. By definition, GLWR should be positively correlated with GL but negatively correlated with GW. As GLWR is mathematically derived from GL and GW, the correlation does not tell too much on genetics. The authors may want to refer the following study for more details: Wang, et al. (2012) On the use of mathematically-derived traits in QTL mapping. *Mol. Breed.* 29: 661–673

(4) Line 93: '1as' should be 'as'

Reviewer #3:

Remarks to the Author:

The authors have adequately addressed all my questions. The manuscript is acceptable now.

Reviewer #4:

Remarks to the Author:

I have gone through the point-by-point response letter to the reviewers and I find that the authors have adequately addressed most of the issues raised by all reviewers (including mine - reviewer #4). Still, some concerns remain:

- I have better understood the model from the author's description in the response letter, but I think, as initially asked, that writing the equation of the model in the method section is required. This is particularly important because the paper is about this new approach. Could you please provide the model equation in the manuscript?

- I am happy that you have carried out a comparison of your predictions to those obtained with a classical GBLUP. Yet, the way GBLUP predictions have been made is not described in the manuscript, and thus I have difficulties evaluating this comparison. Actually, the performances of GBLUP presented in Figure 4 seem very poor, which is unexpected. So I would just make sure that the way the analysis was performed is correct. Could you please give details on the way GBLUP was carried out?

Reviewer #1 (Remarks to the Author):

The manuscript is not improved. Only Figure 2 provides useful results with several QTLs and candidate genes identified. The contents of all the other figures are far from the standard of Nature Communications.

Response: We appreciate your valuable time, expertise in gene fine mapping, and the specific comment on Figure 2. We are grateful that you feel the figure contains useful results with QTL and candidate genes identified.

Reviewer #2 (Remarks to the Author):

I think the authors made a significant improvement in this revision. Most comments from this reviewer have been handled properly. I have only a few minor comments to this revision.

Response: Thank you for the overall comment on the improvement in our revision. Appreciate your additional comments for further improvement.

(1) In abstract, the authors stated that “There were 44 and 97 loci with additive and dominant effects, respectively, including 13 overlaps.” One locus can have both additive (a) and dominant (d) effects. Degree of dominance, i.e. d/a , can take any value. It is not clear how the detected loci were classified to be additive or dominant.

Response: You are correct that the ratio of d/a can take any value. This study conducts statistical tests against null hypothesis that either a or d is zero. However, this study does not have the statistical power to test their ratio to differentiate the levels of dominance relative to additive, such as incomplete dominance, complete dominance, and over dominance. Instead, we aim to identify SNPs with non-zero additive effect or non-zero dominant effects. An SNP with a non-zero additive effect only is defined as an additive locus. An SNP with a non-zero dominant effect only is defined as a dominant locus. An SNP with both a non-zero additive effect and a non-zero dominant effect is defined as an additive and dominant locus. We have revised the method section so that readers can get the details after reading the abstract. Please see method section for part entitled “GWAS pipeline across parents and hybrids for both additive and dominant effects”.

(2) In genetic studies on heterosis in maize or rice, over-dominance was frequently observed. If no over-dominance was observed in this study, the authors may want to discuss the possible reasons.

Response: We added a paragraph to discuss the possible reasons. Please see the last paragraph of the second section in discussion.

(3) I understand that GLWR (i.e. GL/GW) is an important quality index in rice, but the use of a ratio of two measured traits in genetic analysis may be problematic. By definition, GLWR should be positively correlated with GL but negatively correlated with GW. As GLWR is mathematically derived from GL and GW, the correlation does not tell too much on genetics. The authors may want to refer the following study for more details: Wang, et al.

(2012) On the use of mathematically-derived traits in QTL mapping. Mol. Breed. 29: 661–673

Response: Appreciate your analytics and the reference to traits derived mathematically. We added further discussion on the SNPs associated with GL/GW ratio. The recommended reference was cited in the manuscript.

(4) Line 93: '1as' should be 'as'

Response: The mistake is fixed.

Reviewer #3 (Remarks to the Author):

The authors have adequately addressed all my questions. The manuscript is acceptable now.

Response: We appreciate your valuable time and expertise.

Reviewer #4 (Remarks to the Author):

I have gone through the point-by-point response letter to the reviewers and I find that the authors have adequately addressed most of the issues raised by all reviewers (including mine - reviewer #4). Still, some concerns remain:

Response: Thank you for the encouragement to our revision. Appreciate your additional comments for further improvement.

- I have better understood the model from the author's description in the response letter, but I think, as initially asked, that writing the equation of the model in the method section is required. This is particularly important because the paper is about this new approach. Could you please provide the model equation in the manuscript?

Response: We are sorry for our misunderstand in the previous revision. The model equations (1 and 2) are provided in the currently revised manuscript.

- I am happy that you have carried out a comparison of your predictions to those obtained with a classical GBLUP. Yet, the way GBLUP predictions have been made is not described in the manuscript, and thus I have difficulties evaluating this comparison. Actually, the performances of GBLUP presented in Figure 4 seem very poor, which is unexpected. So I would just make sure that the way the analysis was performed is correct. Could you please give details on the way GBLUP was carried out?

Response: Thank you very much for your comments. During adding the description of the classical GBLUP in the method section, we identified a mistake we made previously that caused the unexpected performance of the GBLUP method. We corrected the mistake and re-examined all the analyses. The details of the GBLUP method are included in the method section.

Reviewers' Comments:

Reviewer #2:

Remarks to the Author:

The authors made a significant improvement in this revision. My comments to previous submissions have been handled properly. It can be accepted for publication.

Reviewer #4:

Remarks to the Author:

Thanks for having addressed all my questions (reviewer 4). I think that the manuscript has greatly improved.

Because the performance of gBLUP is not as bad as initially found, I would just suggest removing the word "dramatically" at line 270 in the sentence: "Pyramiding genetic loci for the 12 traits with additive and/or dominant effects in GS dramatically improved the accuracy of predicting hybrid performance, especially compared with the commonly used method, genomic best linear unbiased prediction (gBLUP)".

Reviewer #2 (Remarks to the Author):

The authors made a significant improvement in this revision. My comments to previous submissions have been handled properly. It can be accepted for publication.

Response: We appreciate your valuable time and all the comments for the improvement of our work.

Reviewer #4 (Remarks to the Author):

Thanks for having addressed all my questions (reviewer 4). I think that the manuscript has greatly improved.

Because the performance of gBLUP is not as bad as initially found, I would just suggest removing the word "dramatically" at line 270 in the sentence: "Pyramiding genetic loci for the 12 traits with additive and/or dominant effects in GS dramatically improved the accuracy of predicting hybrid performance, especially compared with the commonly used method, genomic best linear unbiased prediction (gBLUP)".

Response: Yes, thank you for the comments. We removed the word "dramatically" at line 270 according to your advice.